# On the Robustness of Kolmogorov-Arnold Networks: An Adversarial Perspective

**Tal Alter**[*]                                                                 *talalte@post.bgu.ac.il*
*Ben-Gurion University of the Negev, Beer-Sheva, 8410501, Israel*

**Raz Lapid**[*]                                                                 *raz.lapid@deepkeep.ai*
*Ben-Gurion University of the Negev, Beer-Sheva, 8410501, Israel & DeepKeep, Tel-Aviv, Israel*

**Moshe Sipper**                                                                 *sipper@bgu.ac.il*
*Ben-Gurion University of the Negev, Beer-Sheva, 8410501, Israel*

**Reviewed on OpenReview:** *https://openreview.net/forum?id=uafxqhImPM*

## Abstract

Kolmogorov-Arnold Networks (KANs) have recently emerged as a novel paradigm for function approximation by leveraging univariate spline-based decompositions inspired by the Kolmogorov–Arnold theorem. Despite their theoretical appeal—particularly the potential for inducing smoother decision boundaries and lower effective Lipschitz constants—their adversarial robustness remains largely unexplored. In this work, we conduct the first comprehensive evaluation of KAN robustness in adversarial settings, focusing on both fully connected (FCKANs) and convolutional (CKANs) instantiations for image classification tasks. Across a wide range of benchmark datasets (MNIST, FashionMNIST, KMNIST, CIFAR-10, SVHN, and a subset of ImageNet), we compare KANs against conventional architectures using an extensive suite of attacks, including white-box methods (FGSM, PGD, C&W, MIM), black-box approaches (Square Attack, SimBA, NES), and ensemble attacks (AutoAttack). Our experiments reveal that while small- and medium-scale KANs are not consistently more robust than their standard counterparts, large-scale KANs exhibit markedly enhanced resilience against adversarial perturbations. An ablation study further demonstrates that critical hyperparameters—such as number of knots and spline order—significantly influence robustness. Moreover, adversarial training experiments confirm the inherent safety advantages of KAN-based architectures. Overall, our findings provide novel insights into the adversarial behavior of KANs and lay a rigorous foundation for future research on robust, interpretable network designs.

## 1 Introduction

In the rapidly evolving field of deep learning, the robustness of neural networks against adversarial attacks has emerged as a cornerstone of research, driven by the ubiquity of their application, often in sensitive areas. As a result, a multitude of methods have been developed to detect and mitigate adversarial attacks, underscoring the critical need for robust defenses in real-world scenarios. These methods span several areas, including adversarial training (Madry et al., 2017; Bai et al., 2021; Shafahi et al., 2019; Andriushchenko & Flammarion, 2020), defensive distillation (Papernot et al., 2016; Papernot & McDaniel, 2016; Carlini & Wagner, 2016), feature squeezing (Xu et al., 2017), input transformations (Guo et al., 2017; Harder et al., 2021), and using auxiliary models for detection purposes (Zheng & Hong, 2018; Pinhasov et al., 2024; Lapid et al., 2024a). The proliferation of these techniques reflects the ongoing efforts within the research community to fortify neural networks against increasingly sophisticated attacks.

---

[*]Equal contribution

Fully connected neural networks (FCNNs) and convolutional neural networks (CNNs)—foundational models in deep learning—have been extensively studied for their vulnerability and defense mechanisms against adversarial attacks (Dey et al., 2017; Wang et al., 2020; Mohammed et al., 2020; Kerlirzin & Vallet, 1993; Kalina et al., 2022). Recently, the introduction of Kolmogorov-Arnold Networks (KANs) has opened new avenues in function approximation, with theoretical promises of enhanced performance and efficiency (Liu et al., 2024; Bozorgasl & Chen, 2024; Genet & Inzirillo, 2024; Vaca-Rubio et al., 2024; Kiamari et al., 2024).

Many studies have compared the performance of KANs across a range of tasks. For example, Yu et al. (2024) found that FCNNs generally perform better in areas such as machine learning and computer vision. Conversely, KANs excel in representing symbolic formulas.

Another interesting comparison was done by Zeng et al. (2024), focusing on the performance of KANs and FCNNs in handling irregular or noisy functions. They found that KANs exhibited superior performance both for regular and irregular functions. However, in cases involving functions with jump discontinuities or singularities, FCNNs demonstrated greater efficacy.

Dong et al. (2024) studied KANs for time-series classification. They concluded that KANs exhibit significant robustness advantages, attributed to their lower Lipschitz constants. Note that they focused on time-series problems, while we focus herein on image-related tasks; further, they deployed a single attack algorithm (Projected Gradient Descent, PGD; Madry et al. (2017)), whereas we will deploy seven attack algorithms.

This paper delves into the robustness of KANs when faced with adversarial attacks under white-box and black-box conditions. Using multiple datasets—MNIST, KMNIST, FashionMNIST, CIFAR-10, SVHN, and a subset of ImageNet—our analysis not only sheds light on the robustness of KANs but also contributes to the broader understanding of neural network safety in adversarial settings.

The next section provides a comprehensive overview of the architectures employed in this study, along with a detailed discussion of their robustness. In Section 3 we describe the experimental setup and evaluation protocols used to assess our proposed approach. Section 4 then presents the experimental results, reporting how the different architectures perform under a variety of adversarial attacks. Finally, we conclude our study and outline future research directions in Section 5.

## 2 Background

### 2.1 Model Architectures

Our study evaluates two primary types of layers used in artificial neural networks: fully connected layers and convolutional layers. Fully connected layers are the foundational building blocks of feedforward neural networks, where each neuron is connected to every neuron in the next layer. FCNNs are classical feedforward networks that learn floating-point weights for these connections (Cybenko, 1989). In contrast, FCKANs, inspired by the Kolmogorov-Arnold theorem (Kolmogorov, 1961), decompose functions into simpler univariate components, learning these components instead of conventional weights, as detailed by Liu et al. (2024).

Convolutional layers extend the capabilities of fully connected layers by leveraging spatial hierarchies in data such as images. These layers are the first and main layers introduced in CNNs (LeCun et al., 1998), where they typically learn floating-point weights for their convolutional filters. CKANs (Drokin, 2024) apply their unique decomposition principle to learn univariate functions for the filters rather than conventional weights. For more detailed descriptions, please refer to the cited literature.

### 2.2 Robustness in Neural Networks

Robustness in neural networks refers to their ability to maintain performance when subjected to various forms of perturbations, such as noise, adversarial attacks, or distributional shifts. A robust neural network should not only perform well on the training and validation datasets but also generalize effectively to unseen data, including those with slight modifications or corruptions.

Mathematically, robustness can be defined as follows: Consider a neural network $f_\theta : \mathbb{R}^n \to \mathbb{R}^m$, parameterized by $\theta$, and let $\mathbf{x} \in \mathbb{R}^n$ be an input to the network. The network is said to be robust to perturbations if, for a small perturbation, $\delta \in \mathbb{R}^n$, the output of the network remains unchanged, that is:

$$\|f_\theta(\mathbf{x}) - f_\theta(\mathbf{x} + \delta)\| \leq \epsilon \,,$$

for some small $\epsilon > 0$ and for all $\|\delta\| \leq \eta$, where $\eta$ represents the maximum allowable perturbation norm.

Adversarial robustness in particular has gained significant attention in recent years. Adversarial attacks involve making small—often imperceptible—perturbations to input data that can cause a neural network to produce incorrect outputs with high confidence (Goodfellow et al., 2014). These attacks exploit the vulnerability of neural networks to specific input manipulations, raising concerns about their reliability in real-world applications (Szegedy et al., 2013).

For example, the Fast Gradient Sign Method (FGSM) demonstrates how slight changes to image pixels can deceive a classifier into misclassification (Goodfellow et al., 2014). Additionally, the Carlini-Wagner (C&W) attack shows that even robust defenses can be circumvented through optimized perturbations (Carlini & Wagner, 2017).

Such examples underscore the critical need for developing networks with enhanced robustness against adversarial attacks. Numerous other attack methodologies further highlight this necessity, revealing the extensive landscape of adversarial vulnerability in neural networks (Goodfellow et al., 2014; Lapid et al., 2024b; Szegedy et al., 2013; Tamam et al., 2023; Wei et al., 2022; Lapid et al., 2022; Chen et al., 2024; Carlini & Wagner, 2017; Lapid & Sipper, 2023; Andriushchenko et al., 2020; Lapid et al., 2024c).

Several techniques have been proposed to enhance the robustness of neural networks, including adversarial training, regularization methods, and architectural modifications. Adversarial training involves training the network on adversarially perturbed examples, thereby improving its ability to withstand attacks (Madry et al., 2017). Regularization methods—such as dropout (Srivastava et al., 2014) and weight decay—help prevent overfitting and improve generalization. Architectural modifications—such as incorporating attention mechanisms (Vaswani et al., 2017) and leveraging alternative network structures—also contribute to robustness.

The upshot is that robustness is a critical aspect of neural network performance, particularly in the context of adversarial attacks and other perturbations. By understanding and improving the robustness of neural networks, we can develop more reliable and secure models for various applications.

### 2.3 Theoretical Motivation for KAN

Although the primary thrust of our work is empirical, there are several theoretical underpinnings suggesting why KANs may enjoy enhanced adversarial robustness compared to classical neural architectures. In particular, two concepts are worthy of discussion: (1) the functional decomposition principle from Kolmogorov–Arnold theory, and (2) connections to Lipschitz constants and their role in adversarial vulnerability.

### 2.3.1 Kolmogorov-Arnold Function Decomposition

KANs draw inspiration from the Kolmogorov Superposition Theorem (Kolmogorov, 1961; Arnold, 2009), which states that any continuous multivariate function can be decomposed into a finite sum of compositions of continuous univariate functions. In a neural-network instantiation, this translates to learning univariate spline-based transformations, followed by mixing via simpler aggregations. Concretely, one approximates a target function $f \colon \mathbb{R}^n \to \mathbb{R}^m$ by expressing it as a sum or composition of univariate mappings (often denoted $\phi_i$ and $\psi_i$).

Such a decomposition imposes inherent structure on the learned function. Instead of learning a fully unconstrained, high-dimensional parameter space, KANs learn univariate blocks (or "knots") that are each individually simpler. This structure may yield smoother learned decision boundaries, potentially mitigating the local sensitivity that conventional fully connected or convolutional layers can exhibit (Cisse et al., 2017; Ross & Doshi-Velez, 2018).

### 2.3.2 Lipschitz Perspective and Adversarial Vulnerability

A prominent explanation for adversarial vulnerability involves the Lipschitz constant of the model's decision function: networks with larger Lipschitz constants are typically more sensitive to small local perturbations (Cisse et al., 2017; Goodfellow et al., 2014). Formally, a function $f$ is $L$-Lipschitz if:

$$\|f(x) - f(x')\| \le L \cdot \|x - x'\| \quad \text{for all } x, x' \in \mathbb{R}^n. \tag{1}$$

When $L$ is large, small input perturbations can lead to disproportionally large changes in the output (Goodfellow et al., 2014), facilitating adversarial attacks.

In KANs, the piecewise or univariate spline decomposition can, in principle, distribute the function's complexity across multiple univariate components. This might help control or lower effective Lipschitz constants, as each univariate block need not learn steep or extreme gradients. Instead, the overall function is composed of smaller, potentially smoother parts. Empirically, this could translate into fewer local minimas where tiny perturbations trigger large output changes.

## 2.4 Adversarial Attack Types

Adversarial attacks are techniques used to evaluate the robustness of neural networks by introducing small perturbations to input data, causing the model to misclassify the input (Szegedy et al., 2013; Goodfellow et al., 2014). These attacks are broadly categorized into white-box attacks and black-box attacks, based on the adversary's level of access to the model.

**White-Box Attacks**. In white-box attacks, the adversary has full knowledge of the target model, including its architecture, weights, and gradients (Goodfellow et al., 2014; Madry et al., 2017). This access allows the adversary to compute perturbations directly by exploiting the model's vulnerabilities. White-box attacks represent the strongest adversarial threat model, as they assume complete transparency of the system.

**Black-Box Attacks**. In black-box attacks, the adversary has no direct access to the model's internal parameters or gradients (Ilyas et al., 2018; Tu et al., 2019). Instead, the adversary generates adversarial examples by querying the model or leveraging transferability across architectures (Szegedy et al., 2013). Black-box attacks are more practical in real-world scenarios, as they mimic situations where the adversary does not have access to proprietary models or data.

**Ensemble-Based Attacks**. In addition to individual white-box and black-box attacks, ensemble-based attacks combine multiple adversarial strategies to evaluate model robustness more reliably (Croce & Hein, 2020).

# 3 Experimental Setup

**Do KANs offer enhanced safety compared to classical neural network architectures, when confronted with an adversary?**

To address this question we conduct an experimental evaluation of KANs. While it may appear unconventional to begin with experiments, direct assessment of the safety of KANs is most effectively achieved through targeted attacks and subsequent analyses. We evaluate the robustness of FCKANs, CKANs, FCNNs and CNNs under white-box and black-box adversarial attacks. Both fully connected and convolutional architectures are tested across multiple datasets, and the results are analyzed to compare the robustness of the two architectures.

## 3.1 Classifier Models

We utilized six models for each architecture (fully connected and convolutional), categorized into small, medium, and large configurations. The configurations for fully connected models are detailed in Table 1, while those for convolutional models are provided in Table 2. All the KAN models were configured with a uniform setting of *num knots* = 5 and *spline order* = 3.

Table 1: Model configurations for fully connected architectures.

| Model | #Params | Hidden Layers |
|---|---|---|
| FCKAN$_{small}$ | 508,160 | [64] |
| FCNN$_{small}$ | 508,810 | [640] |
| FCKAN$_{medium}$ | 4,730,880 | [256, 1024] |
| FCNN$_{medium}$ | 5,043,210 | [1024, 4096] |
| FCKAN$_{large}$ | 35,676,160 | [128, 128, 256, 256, 256, 512, 512, 512, 1024, 1024, 1024] |
| FCNN$_{large}$ | 33,678,986 | [128, 256, 256, 512, 512, 1024, 1024, 2048, 2048, 4096, 4096] |

Table 2: Model configurations for convolutional architectures. All convolutional layers use a kernel size of $3 \times 3$, stride of 1, and padding of 1. Max pooling is applied after the last convolutional layer, followed by two fully connected layers with hidden dimensions [1024, 512].

(a) Grayscale Datasets

| Model | Output Channels | #Params |
|---|---|---|
| CKAN$_{small}$ | [32, 64, 128] | 27,056,173 |
| CNN$_{small}$ | [32, 64, 128] | 26,316,810 |
| CKAN$_{medium}$ | [64, 64, 64, 128, 128, 128, 256] | 58,554,961 |
| CNN$_{medium}$ | [64, 64, 128, 128, 256, 256, 256] | 53,648,586 |
| CKAN$_{large}$ | [64, 64, 64, 128, 128, 128, 256, 512] | 120,552,018 |
| CNN$_{large}$ | [64, 128, 256, 256, 256, 512, 512, 512] | 110,744,074 |

(b) Multi-Channel Datasets

| Model | Output Channels | #Params |
|---|---|---|
| CKAN$_{small}$ | [32, 64, 128] | 34,925,677 |
| CNN$_{small}$ | [32, 64, 128] | 34,181,706 |
| CKAN$_{medium}$ | [64, 64, 64, 128, 128, 128, 256] | 74,293,969 |
| CNN$_{medium}$ | [64, 64, 128, 128, 256, 256, 256] | 69,378,378 |
| CKAN$_{large}$ | [64, 64, 64, 128, 128, 128, 256, 512] | 152,019,666 |
| CNN$_{large}$ | [64, 128, 256, 256, 256, 512, 512, 512] | 142,202,506 |

All the models were trained using the `AdamW` (Loshchilov & Hutter, 2017) optimizer for 20 epochs. The fully connected models were trained on the MNIST (Deng, 2012), FashionMNIST (Xiao et al., 2017), and KMNIST (Prabhu, 2019) datasets with a learning rate of $1 \times 10^{-4}$, weight decay of $5 \times 10^{-4}$, and a batch size of 64. The convolutional models were trained on the MNIST, SVHN (Netzer et al., 2011), and CIFAR-10 (Krizhevsky, 2009) datasets using a learning rate of $1 \times 10^{-4}$, weight decay of $1 \times 10^{-4}$, and a batch size of 32.

In addition to the standard datasets, we trained convolutional architecutres on a 10-image subset of ImageNet (Deng et al., 2009) to evaluate their performance on high-resolution images. Due to memory constraints, the input resolutions varied based on model size: Small CKAN & CNN models trained on $224 \times 224$ images, Medium CKAN & CNN models trained on $160 \times 160$ images, and Large CKAN & CNN models trained on $112 \times 112$ images. The learning rate, optimizer, and weight decay were kept consistent with other convolutional models, but the batch size was reduced to 4.

## 3.2 Adversarial Attacks

We evaluated the robustness of our models against four distinct white-box attacks. The evaluation began with FGSM (Goodfellow et al., 2014), a foundational attack used as an initial test of model resilience. Subsequently, we assessed the models using more advanced iterative attacks, including PGD (Madry et al., 2017), C&W (Carlini & Wagner, 2017), and the Momentum Iterative Method (MIM) (Dong et al., 2018).

Furthermore, we evaluated the robustness of all the models against three different black-box attack methodologies: transfer-based attacks (Section B.2.1), simple black-box attack (SimBA, (Tu et al., 2019)), square attack (Andriushchenko et al., 2020), and natural evolutionary strategies attack (NES, (Ilyas et al., 2018)). We also added AutoAttack (Croce & Hein, 2020), an ansemble of adversarial attacks including PGD, FAB, Square Attack, and a verification-based attack, which provides a comprehensive robustness evaluation. White-box attacks and AutoAttack were evaluated on all test sets, while black-box attacks were evaluated on a random sample of 1,000 images from the corresponding test set. Transferability metrics were calculated using 5,000 images from the test sets.

The hyperparameter configurations for these experiments are provided in Table 3. For the grayscale datasets (MNIST, FashionMNIST, and KMNIST), an $\epsilon$ value of 32/255 was utilized, whereas for the multi-channel

datasets (CIFAR-10, SVHN, and ImageNet), an $\epsilon$ value of $8/255$ was employed. All experiments were conducted with a batch size of 16, except for ImageNet, which was trained with a batch size of 4. Furthermore, in our implementation of AutoAttack, we adhered to the default hyperparameter settings, as specified in the original AutoAttack paper (Croce & Hein, 2020).

Table 3: Hyperparameters used in experiments with white-box attacks, black-box attacks and ensemble-based attacks. $k$: number of iterations; $\alpha$: step size (learning rate) for perturbations; $c$: confidence parameter in the C&W attack; $\mu$: momentum factor in MIM; $\sigma$: standard deviation of the noise in NES; $n$: number of samples used in NES to estimate gradients. For more details see Appendix B.

Table 4: Ablation study configurations: Varying number of knots and spline order.

| Config | # Knots | Spline Order |
|--------|---------|--------------|
| A | 3 | 5 |
| B | 3 | 7 |
| C | 5 | 5 |
| D | 5 | 3 |
| E | 7 | 3 |

| Attack Type | Attack | Params (Grayscale Datasets) | Params (Multi-channel Datasets) |
|-------------|--------|------------------------------|----------------------------------|
| White-box | PGD | $k = 40$, $\alpha = 0.01$ | $k = 30$, $\alpha = 0.01$ |
| | C&W | $k = 40$, $\alpha = 0.01$, $c = 1$ | $k = 30$, $\alpha = 0.01$, $c = 1$ |
| | MIM | $k = 40$, $\alpha = 0.01$, $\mu = 1$ | $k = 30$, $\alpha = 0.01$, $\mu = 1$ |
| Black-box | Square | $k = 3000$ | $k = 2000$ |
| | SimBA | $k = 5000$ | $k = 3000$ |
| | NES | $k = 300$, $\alpha = 0.025$, $\sigma = 0.0625$, $n = 40$ | $k = 200$, $\alpha = 0.01$, $\sigma = 0.01$, $n = 30$ |

### 3.3 Ablation Study: Impact of KAN Hyperparameters on Robustness

To investigate how KAN hyperparameters affect adversarial robustness, we performed an ablation study focusing on two key factors: *number of knots* and *spline order*. These parameters directly influence the expressiveness and smoothness of function approximation in KANs, which may affect their susceptibility to adversarial attacks. We systematically varied these hyperparameters across all sizes of FCKAN models to assess their impact on adversarial robustness. Each model was trained and evaluated using the same attack settings as described in Section 3.2. The five configurations tested are delineated in Table 4.

### 3.4 Adversarial training

To enhance model robustness, we conducted adversarial training using PGD attack. Adversarial training is a defense strategy where models are trained on adversarially perturbed examples to improve their resilience against future attacks. We applied adversarial training to all the sizes of both fully connected and conv architectures, selecting different datasets for each. The FCKANs and FCNNs were trained on KMNIST data set. The CKANs and CNNs were trained on the CIFAR-10 dataset. Each model was trained for 20 epochs, using a batch size of 16. To asses the effectiveness of adversarial training, we evaluated the trained models against all the white-box and black-box attacks listed in Table 3.

## 4 Results

### 4.1 Fully Connected Models

In the context of iterative white-box adversarial attack methods (PGD, C&W, MIM), small and medium FCKANs demonstrated lower robustness compared to small and medium FCNNs, as shown in Table 5. Across all examined datasets, small and medium FCKANs were notably vulnerable to these attacks. In contrast, small and medium FCNNs demonstrated a measurable—albeit limited—degree of robustness. Nonetheless, even these models remain highly susceptible to adversarial exploitation, rendering them far from secure in practice.

This trend, in which small and medium FCNNs generally outperform FCKANs in robustness under iterative white-box attacks, and for which FCKANs often failed to classify any adversarial images generated by these methods (except for the small FCKAN on MNIST), begins to shift when considering larger configurations. For large models, the robustness of FCKANs improves significantly, surpassing that of large FCNNs under the C&W attack across all datasets. Not only do large FCKANs outperform large FCNNs under the C&W attack, but they also show substantial improvement over their smaller and medium-sized counterparts. Across all iterative adversarial attacks and datasets, large FCKANs exhibited considerably greater resilience than

Table 5: White-box attacks, black-box attacks, and ensemble attacks on fully connected models, evaluated across MNIST, FashionMNIST, and KMNIST datasets. Robust accuracy is given for each corresponding attack. The "Acc" column refers to the clean accuracy of the model. Boldface values indicate the most robust model in each experiment.

| Dataset | Model | White-Box | | | | Black-Box | | | Ensemble-Based Auto Attack | Acc |
|---|---|---|---|---|---|---|---|---|---|---|
| | | FGSM | PGD | C&W | MIM | Square | SimBA | NES | | |
| MNIST | FCKAN$_{small}$ | **6.96** | 0.00 | 0.01 | 0.00 | 0.09 | 0.00 | 0.00 | 0.00 | 96.02 |
| | FCNN$_{small}$ | 1.40 | **0.52** | **0.65** | **0.62** | **2.38** | **0.19** | **2.08** | **0.30** | 97.94 |
| | FCKAN$_{medium}$ | 9.51 | 0.00 | 0.00 | 0.00 | 0.00 | 0.19 | 0.00 | 0.00 | 97.62 |
| | FCNN$_{medium}$ | **10.80** | **0.77** | **0.87** | **1.05** | **3.27** | **18.35** | **19.94** | **0.25** | 98.31 |
| | FCKAN$_{large}$ | **37.83** | 1.64 | **2.27** | 0.42 | **24.20** | **38.78** | 0.00 | 0.00 | 94.54 |
| | FCNN$_{large}$ | 9.57 | **4.78** | 0.15 | **4.76** | 2.38 | 5.75 | **5.25** | **0.05** | 97.47 |
| FashionMNIST | FCKAN$_{small}$ | **4.84** | 0.00 | 0.00 | 0.00 | 0.00 | 0.00 | 0.00 | 0.00 | 86.83 |
| | FCNN$_{small}$ | 0.95 | **0.23** | **0.44** | **0.32** | **3.86** | **0.39** | **3.67** | **0.14** | 88.20 |
| | FCKAN$_{medium}$ | 3.78 | 0.00 | 0.00 | 0.00 | 0.00 | 0.00 | 0.00 | 0.00 | 86.72 |
| | FCNN$_{medium}$ | **5.23** | **0.12** | **0.11** | **0.18** | **1.58** | **11.30** | **11.80** | **0.01** | 89.14 |
| | FCKAN$_{large}$ | **16.68** | 0.05 | **0.09** | 0.00 | **19.64** | **29.06** | 0.00 | **0.00** | 83.24 |
| | FCNN$_{large}$ | 1.48 | **0.11** | 0.00 | **0.11** | 0.19 | 2.48 | **2.67** | **0.00** | 88.53 |
| KMNIST | FCKAN$_{small}$ | 2.80 | 0.00 | 0.00 | 0.00 | 0.00 | 0.00 | 0.00 | 0.00 | 81.71 |
| | FCNN$_{small}$ | **3.09** | **1.16** | **1.52** | **1.41** | **7.24** | **0.89** | **5.55** | **0.90** | 90.28 |
| | FCKAN$_{medium}$ | 7.65 | 0.00 | 0.00 | 0.00 | 0.00 | 1.48 | 1.68 | 0.00 | 87.47 |
| | FCNN$_{medium}$ | **11.74** | **1.01** | **1.11** | **1.35** | **6.54** | **8.82** | **17.85** | **0.37** | 91.15 |
| | FCKAN$_{large}$ | **25.90** | 0.36 | **0.53** | 0.04 | **14.78** | **23.61** | 0.00 | 0.00 | 78.92 |
| | FCNN$_{large}$ | 7.14 | **0.70** | 0.29 | **1.05** | 3.86 | 1.38 | **2.67** | **0.06** | 89.28 |

small and medium FCKANs. In contrast, large FCNNs often demonstrated lower robustness than their smaller variants.

Further evidence of the robustness of large FCKANs under white-box iterative attacks is shown in Figure 1. The figure illustrates that the loss function, which increases as the discrepancy between the predicted and true labels is maximized, remains consistently lower for large FCKANs compared to other models. This indicates that deceiving large FCKANs is more challenging. Additionally, while the losses of other models tend to converge during attacks, the loss of large FCKANs maintains a linear trajectory, further underscoring their resilience.

A particularly notable observation is the superior performance of large FCKANs compared to large FCNNs under the FGSM white-box adversarial attack, with a pronounced gap favoring FCKANs across all datasets. This is further supported by Figure 2, which illustrates the clear advantage of large FCKANs. While the differences in robustness between small and medium FCNNs and FCKANs are less definitive, the superiority of large FCKANs over large FCNNs is both clear and substantial, firmly establishing FCKANs as the more robust architecture in larger configurations.

When considering black-box attacks (Square, SimBA, and NES), the performance dynamics shift. Small FCNNs outperform FCKANs on all datasets, where small FCKANs failed to classify any image generated with those attacks. When considering the medium models, although FCKANs successfully classified a small number of images (especially on the KMNIST dataset), medium FCNNs still performed better than medium FCKANs across all datasets. However, for larger models, FCKANs demonstrate marked improvement in robustness against SimBA and Square black-box attacks, significantly surpassing FCNNs. In contrast, similar to the small and medium sizes under the NES attack, FCKANs failed to classify any adversarial image. The fact that FCKANs failed under the NES attack could result from the gradient approximation for these architectures being much more accurate than for the FCNN architecture.

All the FCNN models, except the large FCNN on FashionMNIST, successfully classify at least one image generated with AutoAttack, while none of the FCKAN models, across all datasets, successfully classify an image generated with that attack. This may be due to the structural properties of FCKANs, which, while beneficial for robustness against certain attacks, might also introduce vulnerabilities that AutoAttack

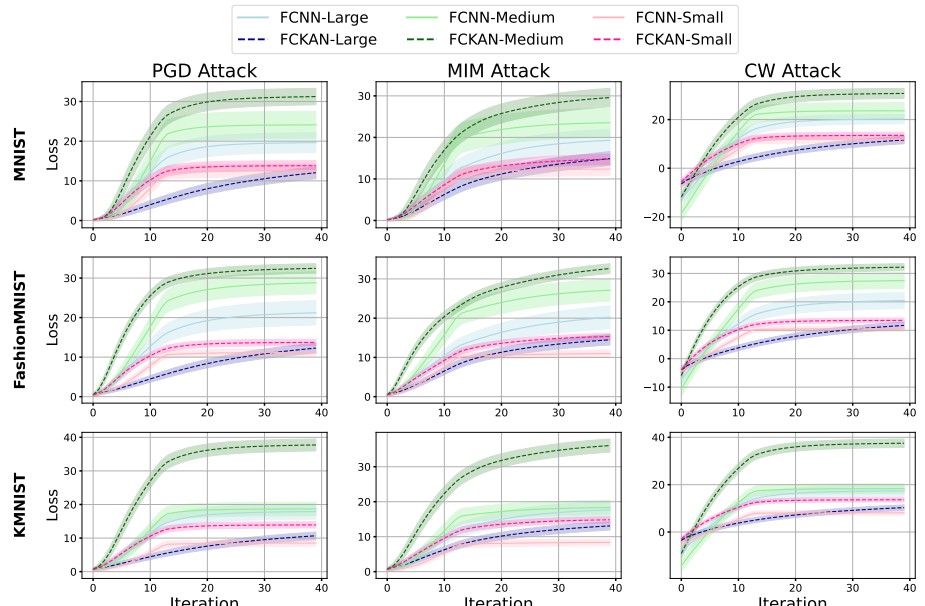

Figure 1: Comparative loss dynamics of FCNNs and FCKANs under PGD, MIM, and C&W attacks on the MNIST, FashionMNIST, and KMNIST datasets (from top to bottom, respectively). Each line represents the mean loss across batches, per iteration, with shaded areas indicating the standard deviation.

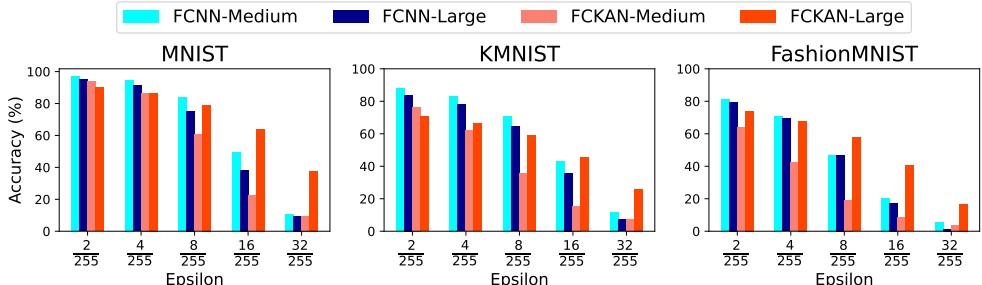

Figure 2: Robust accuracy of FCNN$_{medium}$, FCNN$_{large}$, FCKAN$_{medium}$, and FCKAN$_{large}$ as a function of FGSM attack strength, with varying $\epsilon$ values. Each bar represents a different model, and robustness is measured as the accuracy of the model against adversarial examples generated with specific $\epsilon$ values. The x-axis indicates the $\epsilon$ values used for FGSM attacks, while the y-axis shows the corresponding robust accuracy of the models.

effectively exploits. Specifically, AutoAttack's adaptive nature could be particularly well-suited to finding perturbations that significantly degrade FCKAN performance.

In our analysis of the transferability of adversarial white-box attacks (Table 6) , we observed that across all datasets, the large FCKAN consistently exhibited the lowest percentage of misclassified images among all six models. This suggests that fooling the large FCKAN is the most challenging task. Moreover, except for the small and medium models on the KMNIST dataset, FCKANs consistently demonstrated greater resistance to adversarial attacks than their FCNN counterparts for each model size (small, medium and large), highlighting the superior robustness of FCKANs at every scale.

Qualitative results corresponding to these attacks are presented in Appendix A.

## 4.2 Convolutional Models

In the context of iterative white-box adversarial attacks, CKANs and CNNs exhibit contrasting behaviors across datasets. As observed in Table 7, both CKANs and CNNs exhibit poor performance on the multi-

Table 6: Maximum transferability of our fully connected models, for FashionMNIST, KMNIST, and MNIST. Each row represents the model that generates the adversarial example and each column represents the model that evaluates those examples. The value in row $i$, column $j$ represents the maximum transferability between row-$i$ model and column-$j$ model across four attacks—FGSM, PGD, C&W, MIM—calculated using Equation 12. Unlike robustness tables, the values here indicate the percentage of images misclassified. The 'Average' row shows the average transferability for the attacking model. Boldfaced values denote the lowest transferability, where a lower value indicates better robustness.

FashionMNIST

| | $FCKAN_{small}$ | $FCKAN_{medium}$ | $FCKAN_{large}$ | $FCNN_{small}$ | $FCNN_{medium}$ | $FCNN_{large}$ |
|---|---|---|---|---|---|---|
| $FCKAN_{small}$ | - | 57.32 | 38.97 | 59.98 | 56.23 | 48.76 |
| $FCKAN_{medium}$ | 40.55 | - | 28.11 | 39.89 | 39.99 | 35.81 |
| $FCKAN_{large}$ | 32.34 | 29.91 | - | 31.90 | 26.03 | 27.18 |
| $FCNN_{small}$ | 81.34 | 81.57 | 59.43 | - | 98.46 | 90.64 |
| $FCNN_{medium}$ | 64.82 | 68.91 | 45.16 | 90.19 | - | 85.33 |
| $FCNN_{large}$ | 57.13 | 59.31 | 39.45 | 75.18 | 82.21 | - |
| Average ($\downarrow$) | 55.23 | 59.40 | **42.22** | 59.42 | 60.58 | 57.54 |

KMNIST

| | $FCKAN_{small}$ | $FCKAN_{medium}$ | $FCKAN_{large}$ | $FCNN_{small}$ | $FCNN_{medium}$ | $FCNN_{large}$ |
|---|---|---|---|---|---|---|
| $FCKAN_{small}$ | - | 29.58 | 21.11 | 21.60 | 13.77 | 19.71 |
| $FCKAN_{medium}$ | 32.87 | - | 17.47 | 16.19 | 10.94 | 17.43 |
| $FCKAN_{large}$ | 19.59 | 11.82 | - | 7.99 | 4.69 | 11.30 |
| $FCNN_{small}$ | 70.68 | 62.29 | 45.95 | - | 89.77 | 71.58 |
| $FCNN_{medium}$ | 51.04 | 42.42 | 30.76 | 83.49 | - | 59.40 |
| $FCNN_{large}$ | 45.75 | 35.72 | 31.89 | 50.22 | 43.74 | - |
| Average ($\downarrow$) | 43.98 | 36.36 | **29.43** | 35.89 | 32.58 | 35.88 |

MNIST

| | $FCKAN_{small}$ | $FCKAN_{medium}$ | $FCKAN_{large}$ | $FCNN_{small}$ | $FCNN_{medium}$ | $FCNN_{large}$ |
|---|---|---|---|---|---|---|
| $FCKAN_{small}$ | - | 28.10 | 22.94 | 55.10 | 30.55 | 32.32 |
| $FCKAN_{medium}$ | 35.33 | - | 17.71 | 38.83 | 22.67 | 26.74 |
| $FCKAN_{large}$ | 22.78 | 08.94 | - | 18.50 | 08.36 | 14.56 |
| $FCNN_{small}$ | 81.35 | 68.48 | 48.68 | - | 95.84 | 88.31 |
| $FCNN_{medium}$ | 64.17 | 54.40 | 33.83 | 95.48 | - | 84.39 |
| $FCNN_{large}$ | 58.89 | 44.78 | 31.77 | 78.42 | 72.29 | - |
| Average ($\downarrow$) | 52.50 | 40.94 | **30.98** | 57.26 | 45.94 | 49.26 |

channel datasets (CIFAR-10, SVHN and ImageNet), demonstrating significant vulnerability to adversarial perturbations. On CIFAR-10, none of the models successfully classify any images generated by those attacks. On the SVHN dataset, the medium and large models manage to classify a small number of adversarially generated images. For ImageNet, the CKAN models failed to classify any images, exhibiting the same behavior observed on CIFAR-10. However, CNNs on medium and large scales did not fail completely, showing some level of robustness compared to their CKAN coutnerparts.

Conversely, for the simple dataset (MNIST), CKANs exhibit significantly better robustness than CNNs across all model sizes. Notably, medium and large CKAN models demonstrate remarkable resilience, successfully classifying a significant proportion of adversarial images. In contrast, CNNs perform poorly under these conditions, managing to classify only a negligible number of adversarial images.

This disparity is further corroborated by Figure 3, which reveals a faster loss convergence on complex datasets (CIFAR-10, SVHN and ImageNet) compared to MNIST during the attack process. The rapid convergence of the loss implies that adversarial attacks "fool" models more easily on these complex datasets.

For the non-iterative white-box attack (FGSM), the results are more definitive. On CIFAR-10, CKANs exhibit greater robustness than CNNs for medium-sized models, whereas CNNs perform better for large

Table 7: White-box, black-box, and ensemble attack on convolutional models, evaluated across MNIST, CIFAR-10, SVHN, and ImageNet datasets. The table reports robust accuracy for each attack, while the Acc column indicates the model's accuracy on clean, unperturbed inputs. Boldface values indicate the most robust model in each experiment. (For ImageNet NES attacks, denoted by ∗, results could not be fully evaluated as in Ilyas et al. (2018) due to computational constraints at our university.)

| Dataset | Model | White-Box | | | | Black-Box | | | Ensemble-Based Auto Attack | Acc |
|---|---|---|---|---|---|---|---|---|---|---|
| | | FGSM | PGD | C&W | MIM | Square | SimBA | NES | | |
| MNIST | $CKAN_{small}$ | **72.36** | **4.52** | **5.70** | **8.98** | **38.39** | **34.82** | **49.90** | **1.13** | 99.47 |
| | $CNN_{small}$ | 68.20 | 0.74 | 0.98 | 2.26 | 27.38 | 9.82 | 33.73 | 0.02 | 99.23 |
| | $CKAN_{medium}$ | **86.33** | **29.88** | **33.12** | **42.72** | **57.93** | **41.76** | **66.07** | **14.78** | 99.55 |
| | $CNN_{medium}$ | 28.36 | 0.00 | 0.00 | 0.00 | 0.39 | 7.24 | 8.92 | 0.00 | 99.36 |
| | $CKAN_{large}$ | **84.72** | **31.28** | **34.32** | **39.64** | **56.84** | **36.21** | **58.92** | **18.89** | 99.52 |
| | $CNN_{large}$ | 55.87 | 0.01 | 0.13 | 0.19 | 5.15 | 2.38 | 5.45 | 0.00 | 99.49 |
| CIFAR-10 | $CKAN_{small}$ | **0.01** | **0.00** | **0.00** | **0.00** | **0.00** | 2.48 | 0.99 | **0.00** | 74.70 |
| | $CNN_{small}$ | **0.01** | **0.00** | **0.00** | **0.00** | **0.00** | 2.28 | 0.69 | **0.00** | 73.03 |
| | $CKAN_{medium}$ | **0.57** | **0.00** | **0.00** | **0.00** | **0.39** | **4.26** | **1.88** | **0.00** | 80.72 |
| | $CNN_{medium}$ | 0.27 | **0.00** | **0.00** | **0.00** | 0.09 | 2.28 | 0.79 | **0.00** | 76.43 |
| | $CKAN_{large}$ | 0.85 | **0.00** | **0.00** | **0.00** | 0.19 | 4.06 | 1.98 | **0.00** | 82.82 |
| | $CNN_{large}$ | **1.94** | **0.00** | **0.00** | **0.00** | **1.48** | **4.46** | **2.38** | **0.00** | 79.26 |
| SVHN | $CKAN_{small}$ | **1.04** | **0.00** | **0.00** | **0.00** | **7.44** | **10.41** | **7.44** | **0.00** | 92.34 |
| | $CNN_{small}$ | 0.43 | **0.00** | **0.00** | **0.00** | 4.76 | 5.15 | 2.87 | **0.00** | 90.69 |
| | $CKAN_{medium}$ | **6.74** | 0.02 | 0.02 | 0.02 | **16.96** | **16.36** | **13.59** | 0.01 | 94.37 |
| | $CNN_{medium}$ | 5.00 | **0.05** | **0.05** | **0.04** | 14.78 | 9.52 | 8.03 | **0.03** | 93.41 |
| | $CKAN_{large}$ | **8.33** | **0.09** | **0.10** | **0.09** | **19.84** | **16.26** | **13.09** | **0.06** | 94.41 |
| | $CNN_{large}$ | 6.67 | 0.04 | 0.05 | 0.04 | 14.08 | 8.23 | 7.83 | 0.02 | 92.72 |
| ImageNet | $CKAN_{small}$ | 0.20 | **0.00** | **0.00** | **0.00** | **59.40** | 70.60 | 76.20∗ | **0.00** | 76.20 |
| | $CNN_{small}$ | **1.60** | **0.00** | **0.00** | **0.00** | 58.80 | **74.60** | 79.80∗ | **0.00** | 79.80 |
| | $CKAN_{medium}$ | **4.00** | 0.00 | 0.00 | 0.00 | **45.00** | **64.60** | 78.20∗ | 0.00 | 78.20 |
| | $CNN_{medium}$ | 3.00 | **0.20** | **0.20** | **0.20** | 41.80 | 63.60 | 77.40∗ | **0.20** | 77.40 |
| | $CKAN_{large}$ | 6.40 | 0.00 | 0.00 | 0.00 | 22.20 | 37.20 | 72.80∗ | **0.80** | 72.80 |
| | $CNN_{large}$ | **11.20** | **0.80** | **1.20** | **1.00** | **34.80** | **47.00** | 73.40∗ | 0.60 | 73.40 |

models. On ImageNet, small and large CNNs achieve greater robustness than CKANs. However, similar to CIFAR-10, the medium CKAN model demonstrates greater robustness than its CNN counterpart. On the SVHN dataset, CKANs demonstrate slightly higher robustness across all model sizes, although the gap between CKANs and CNNs is relatively small. On the MNIST dataset, which is less complex, CKANs show significantly higher robustness than CNNs, with a noticeable gap favoring CKANs. As shown in Figure 4, this gap is not consistently large across different $\epsilon$ values.

When analyzing black-box attacks, for MNIST, CIFAR-10, and SVHN, CKANs consistently demonstrated superior robustness compared to CNNs across most datasets and model sizes. CKANs outperformed CNNs in nearly all scenarios, except for the black-box attacks on large models for the CIFAR-10 dataset, where CNNs exhibited slightly better robustness, and on the small models for the CIFAR-10 dataset, where both of the configurations failed to classify any adversarial image generated with the Square attack. For ImageNet, all models performed well under black-box attacks. However, for large models, CNNs significantly outperformed CKANs in terms of robustness.

In the case of AutoAttack on the MNIST dataset, CKANs performed exceptionally well against that attack. This suggests that fooling a large CKAN model is particularly challenging when dealing with grayscale images, potentially due to the unique structural properties of CKANs, which enhance their resistance to adversarial perturbations in such domains. In contrast, AutoAttack successfully "fooled" most of the models in the other datasets. On CIFAR-10, none of the models successfully classified even a single adversarial image generated by this attack. On SVHN and ImageNet, the medium CNN outperformed the medium CKAN, while the large CKAN performed better than the large CNN. The small models, similar to the CIFAR-10 models, failed to classify even a single image.

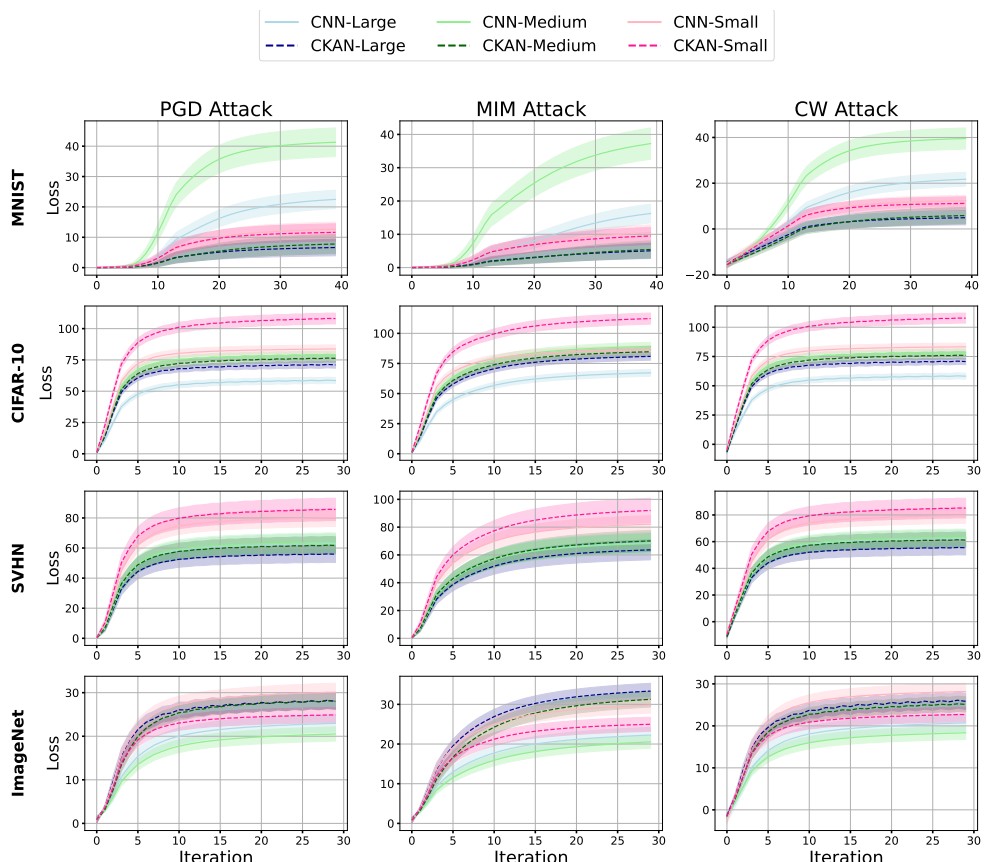

Figure 3: Comparative loss dynamics of CNNs and CKANs under PGD, MIM, and C&W attacks on the MNIST, CIFAR-10, SVHM and ImageNet datasets. Each line represents the mean loss across batches, per iteration, with shaded areas indicating the standard deviation. For the MNIST, CIFAR-10 AND SVHN datasets, the shaded areas represent the original standard deviation, whereas for ImageNet, they represent the logarithm of the original std

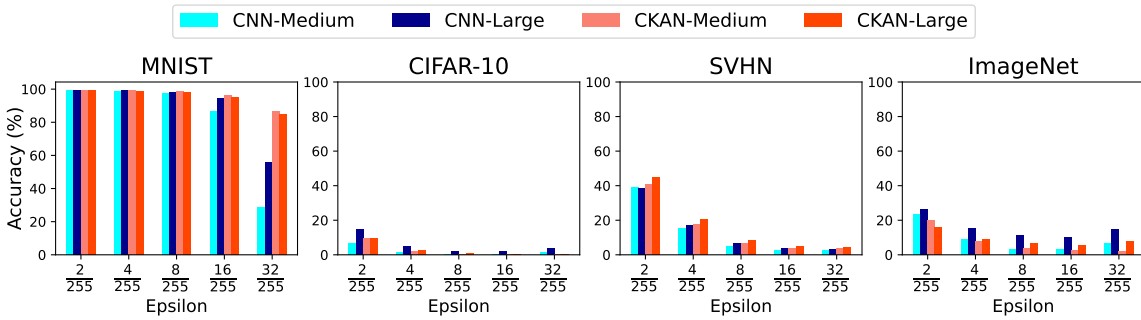

Figure 4: Robust accuracy of $CNN_{medium}$, $CNN_{large}$, $CKAN_{medium}$, and $CKAN_{large}$ as a function of FGSM attack strength, with varying $\epsilon$ values. Each bar represents a different model, and robustness is measured as the accuracy of the model against adversarial examples generated with specific $\epsilon$ values. The x-axis indicates the $\epsilon$ values used for FGSM attacks, while the y-axis shows the corresponding robust accuracy of the models.

In our analysis of the transferability of adversarial white-box attacks (Table 8), we observed that the results vary across datasets and model configurations, with no clear overall trend. On the MNIST dataset, fooling the medium CKAN proves to be the most challenging task. On the SVHN dataset, the large CKAN is the hardest modle to fool. Conversely, on the CIFAR-10 dataset, the large CKAN is the easiest model to fool, right after the medium CNN model. Similar to the transferability results for the fully connected models,

CKANs consistently demonstrated greater resistance to adversarial attacks than their CNN counterparts for each model size, with the exception of the large size on the CIFAR-10 dataset.

Table 8: Maximum transferability of our convolutional models for MNIST, CIFAR-10, and SVHM. Each row represents the model that generates the adversarial example, while each column represents the model that evaluates those examples. The value in row $i$, column $j$ represents the maximum transferability between row-$i$ model and column-$j$ model across four attacks—FGSM, PGD, C&W, MIM—calculated using Equation 12. Unlike robustness tables, the values here indicate the percentage of images misclassified. The 'Average' row shows the average transferability for the attacking model. Boldface values denoting the lowest transferability, where a lower value indicates better robustness.

**MNIST**

|  | $CKAN_{small}$ | $CKAN_{medium}$ | $CKAN_{large}$ | $CNN_{small}$ | $CNN_{medium}$ | $CNN_{large}$ |
|---|---|---|---|---|---|---|
| $CKAN_{small}$ | - | 2.94 | 3.26 | 4.70 | 20.6 | 4.80 |
| $CKAN_{medium}$ | 5.30 | - | 17.78 | 3.32 | 18.95 | 12.01 |
| $CKAN_{large}$ | 3.75 | 14.95 | - | 2.51 | 15.16 | 9.93 |
| $CNN_{small}$ | 3.43 | 1.71 | 1.55 | - | 33.47 | 6.72 |
| $CNN_{medium}$ | 3.82 | 2.78 | 3.48 | 7.80 | - | 46.28 |
| $CNN_{large}$ | 3.35 | 4.49 | 5.28 | 5.64 | 6.98 | - |
| Average ($\downarrow$) | **3.93** | 5.37 | 6.27 | 4.79 | 19.03 | 15.94 |

**CIFAR-10**

|  | $CKAN_{small}$ | $CKAN_{medium}$ | $CKAN_{large}$ | $CNN_{small}$ | $CNN_{medium}$ | $CNN_{large}$ |
|---|---|---|---|---|---|---|
| $CKAN_{small}$ | - | 53.39 | 45.62 | 62.95 | 60.19 | 35.45 |
| $CKAN_{medium}$ | 70.25 | - | 93.48 | 79.12 | 90.40 | 68.52 |
| $CKAN_{large}$ | 62.80 | 92.13 | - | 75.85 | 89.65 | 73.39 |
| $CNN_{small}$ | 70.47 | 71.27 | 69.74 | - | 85.70 | 59.78 |
| $CNN_{medium}$ | 63.23 | 82.31 | 82.52 | 82.50 | - | 78.68 |
| $CNN_{large}$ | 47.53 | 68.93 | 76.86 | 65.07 | 84.74 | - |
| Average ($\downarrow$) | **62.85** | 73.60 | 73.64 | 73.09 | 82.13 | 63.16 |

**SVHN**

|  | $CKAN_{small}$ | $CKAN_{medium}$ | $CKAN_{large}$ | $CNN_{small}$ | $CNN_{medium}$ | $CNN_{large}$ |
|---|---|---|---|---|---|---|
| $CKAN_{small}$ | - | 63.08 | 58.62 | 72.23 | 58.73 | 57.70 |
| $CKAN_{medium}$ | 77.44 | - | 83.76 | 69.97 | 75.40 | 75.40 |
| $CKAN_{large}$ | 76.54 | 86.43 | - | 69.42 | 76.23 | 75.67 |
| $CNN_{small}$ | 68.52 | 55.79 | 52.85 | - | 63.33 | 62.15 |
| $CNN_{medium}$ | 72.11 | 74.21 | 71.98 | 78.26 | - | 85.87 |
| $CNN_{large}$ | 70.93 | 74.52 | 72.70 | 78.62 | 85.89 | - |
| Average ($\downarrow$) | 73.10 | 70.80 | **67.98** | 73.70 | 71.91 | 71.35 |

Our experiments indicate that smaller KANs do not consistently outperform conventional neural networks in adversarial settings. By contrast, once the KAN architecture is scaled, the decomposition advantage becomes more pronounced. These findings cohere with the theoretical conjecture stated in Section 2.3.1, namely, that having sufficient capacity allows KANs to learn smoother local transformations, distributing the learned decision boundary in a way that reduces susceptibility to adversarial perturbations.

While we do not claim a fully rigorous theory of adversarial robustness for KANs, these lines of reasoning provide plausible explanations for our empirical results, motivating further research on KAN-based Lipschitz regularization and other theoretical investigations.

## 4.3 Ablation Study: Impact of KAN Hyperparameters

To evaluate the impact of KAN hyperparameters on adversarial robustness, we analyzed the five configurations tested in our ablation study (Section 3.3).

The results in Figure 5 reveal an interesting phenomenon. In small models, across all datasets, the model with the highest spline order consistently achieves better robustness against adversarial attacks, except for FGSM, where a model with a spline order of 5 performs best. This trend remains stable in medium-sized models, where we observe that the model with the lowest spline order (3) performs significantly worse at correctly predicting adversarial examples.

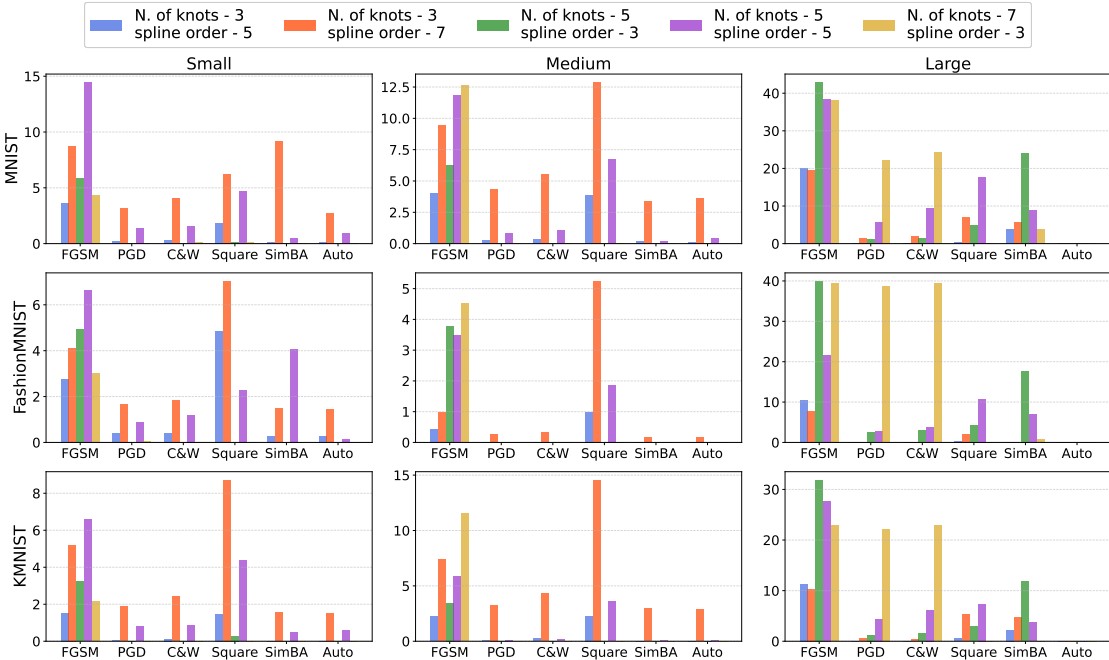

Figure 5: Robust accuracy of FCKANs under different adversarial attacks, varying the number of knots and spline order.

However, when moving to large models, this trend shifts. In this case, models with the lowest spline order demonstrate strong adversarial robustness. Notably, for iterative white-box attacks, the model with 7 knots and a spline order of 3 exhibits remarkable stability, showing strong resistance to these attacks.

### 4.4 Adversarial training

Examining the results in Table 9 for the fully connected models, and Table 10 for the convolutional models, we can clearly see that, although the standard FCNNs and CNNs achieved higher accuracy on clean test-set images (except for the large models), they perform worse than their KAN-bases counterparts (FCKANs and FCKANs) when predicting adversarially perturbed images. This trend holds across all attack types, indicating that FCKAN and CKAN models are inherently more robust to adversarial perturbations compared to traditional neural networks. Furthermore, this improved robustness may stem from the ability of FCKANs and CKANs to better adapt to adversarial perturbations after exposure to adversarial examples during training. Their structured functional representations might enable them to generalize more effectively to unseen adversarial examples compared to FCNNs and CNNs, which struggle to adjust beyond the specific perturbations encountered during training.

## 5 Concluding Remarks

The introduction of KANs opens new avenues in adversarial machine learning, presenting both challenges and opportunities for improving model safety. In this study, we explored several key aspects of KANs, focusing on their behavior in convolutional and fully connected layers, which have not been extensively investigated before.

Table 9: Adversarial attack results on fully connected models trained on KMNIST. Values represent robust accuracy (%) after each attack. The highest robust accuracy per row is boldfaced.

| Model | FGSM | PGD | C&W | MIM | Square | SimBA | NES | Clean Acc |
|---|---|---|---|---|---|---|---|---|
| FCKAN$_{small}$ | **68.5** | **62.14** | **60.87** | **62.41** | **55.75** | **54.16** | **60.61** | 79.24 |
| FCNN$_{small}$ | 54.63 | 49.12 | 47.78 | 49.71 | 52.97 | 47.42 | 57.73 | 85.56 |
| FCKAN$_{medium}$ | **77.12** | **72.38** | **71.73** | **72.85** | **71.03** | **69.44** | **74.20** | 88.95 |
| FCNN$_{medium}$ | 72.73 | 65.95 | 66.60 | 67.33 | 70.63 | 63.69 | 73.01 | 95.22 |
| FCKAN$_{large}$ | **64.72** | **57.82** | **57.21** | **58.55** | **53.17** | **51.88** | **56.54** | 74.87 |
| FCNN$_{large}$ | 48.05 | 41.92 | 41.7 | 43.21 | 46.42 | 40.87 | 50.09 | 74.08 |

Table 10: Adversarial attack results on convolutional models trained on CIFAR-10. Values represent robust accuracy (%) after each attack. The highest robust accuracy per row is boldfaced.

| Model | FGSM | PGD | C&W | MIM | Square | SimBA | NES | Clean Acc |
|---|---|---|---|---|---|---|---|---|
| CKAN$_{small}$ | **35.81** | **31.29** | **29.64** | **31.48** | **50.19** | **35.61** | **51.78** | 64.31 |
| CNN$_{small}$ | 30.28 | 23.68 | 23.80 | 23.96 | 46.62 | 26.88 | 46.82 | 65.97 |
| CKAN$_{medium}$ | **39.58** | **34.61** | **33.44** | **34.96** | **52.28** | **39.58** | **54.06** | 65.80 |
| CNN$_{medium}$ | 28.55 | 21.48 | 21.61 | 21.55 | 44.24 | 27.18 | 44.14 | 66.27 |
| CKAN$_{large}$ | **40.56** | **35.56** | **34.77** | **35.73** | **52.38** | **40.57** | **51.88** | 66.60 |
| CNN$_{large}$ | 31.74 | 25.02 | 24.73 | 25.12 | 45.23 | 28.57 | 47.22 | 66.42 |

First, we assessed the inherent robustness of FCKANs and CKANs against both white-box and black-box adversarial attacks. Our findings indicate that while FCKANs exhibit vulnerability levels similar to traditional FCNNs in smaller configurations, they demonstrate significant improvements in robustness as network size increases. This suggests that FCKANs could be particularly effective in scenarios where larger models are feasible and robustness is a priority. In contrast to the trends observed for FCKANs and FCNNs, CKANs consistently exhibit greater robustness than CNNs of comparable sizes across most attacks. This highlights a distinct advantage of CKANs in adversarial settings, even when network sizes remain constant.

Our ablation study further highlights the impact of key hyperparameters, specifically the number of knots and spline order, on adversarial robustness. We found that models with a higher spline order tend to be more resilient to adversarial perturbations, especially in smaller and medium-sized configurations. However, for large-scale models, lower spline order configurations demonstrated stronger robustness under iterative white-box attacks, suggesting that the interplay between network complexity and attack resilience is non-trivial. These findings indicate that optimizing KAN hyperparameters could be a promising avenue for enhancing robustness.

Moreover, our adversarial training experiments demonstrate that FCKANs and CKANs benefit significantly from exposure to adversarial examples during training. Notably, adversarially trained KAN models consistently outperformed their standard counterparts in robustness across all tested attacks. However, even after adversarial training, the performance gap between KANs and conventional architectures persisted in some settings, suggesting that additional regularization techniques or training strategies could further enhance their resilience.

While these results provide valuable insights, they also raise important open questions. Future work should explore whether alternative training methodologies—such as certified defenses, Lipschitz regularization, or hybrid adversarial training—can further improve KAN robustness.

Despite their theoretical advantages, KANs are not inherently immune to adversarial perturbations. However, their structured functional decomposition offers a unique framework that may be leveraged for more robust and interpretable deep learning models. As adversarial machine learning research advances, continued investigation into the interplay between KAN structure, optimization, and adversarial robustness will be crucial for their practical adoption in security-critical applications.

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

# A    Image Samples of Attacks

This section provides visual examples of adversarial attacks applied to a single MNIST image, demonstrating the unique perturbation patterns across different model architectures and sizes. These visualizations compare FCNNs with FCKANs and CNNs with CKANs under white-box and black-box attack scenarios.

**Fully Connected Models.** Figure 6 and Figure 7 depict adversarial images and their corresponding perturbations for FCNNs and FCKANs under white-box and black-box attacks, respectively. The results highlight distinct perturbation behaviors, underscoring the unique adversarial behaviors of KANs.

**Convolutional Models.** Figure 8 and Figure 9 illustrate adversarial images and perturbations for CNNs and CKANs under white-box and black-box attacks, showcasing architectural influences on adversarial patterns. Similarly, Figure 10 illustrates adversarial images and perturbations for the ImageNet dataset under white-box attacks, showcasing the impact of different architectures on adversarial behavior.

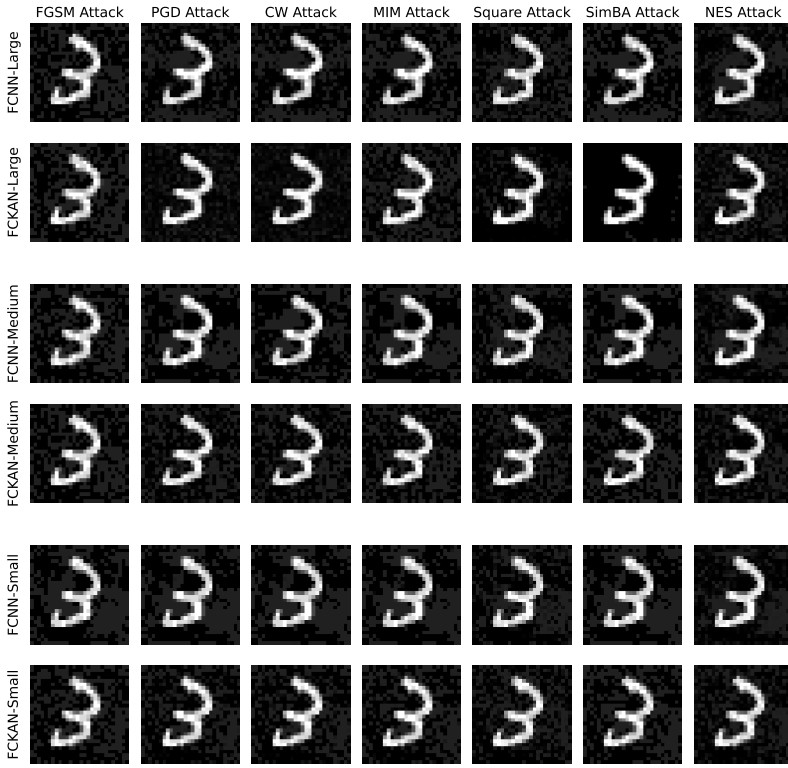

Figure 6: Visualization of the adversarial images on a single MNIST image using fully connected models. Rows represent model types and sizes: FCNN$_\text{large}$, FCKAN$_\text{large}$, FCNN$_\text{medium}$, FCKAN$_\text{medium}$, FCNN$_\text{small}$, and FCKAN$_\text{small}$.

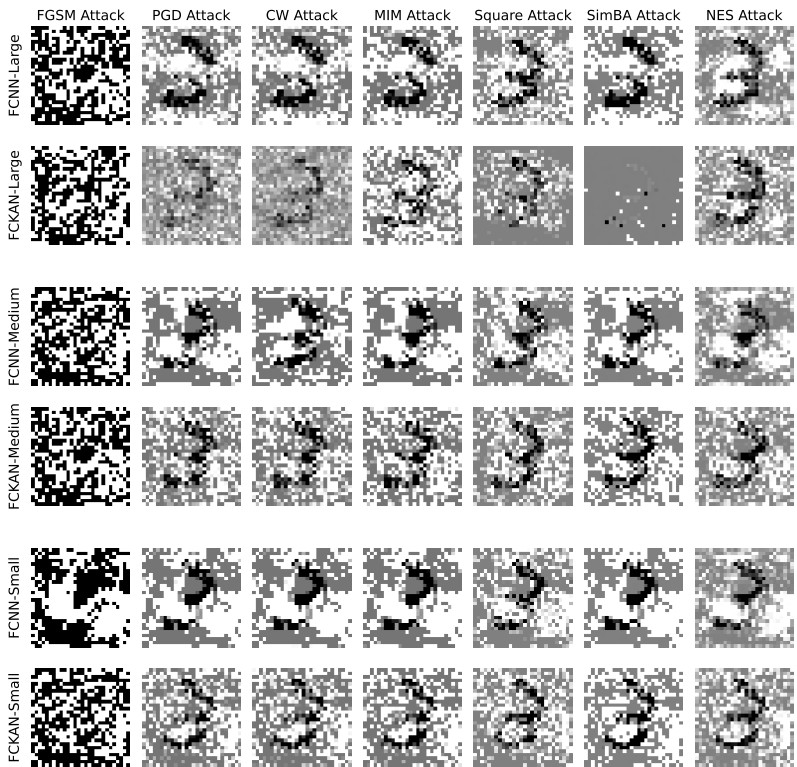

Figure 7: Visualization of the pertubations on a single MNIST image using fully connected models. Rows represent model types and sizes: $\text{FCNN}_{large}$, $\text{FCKAN}_{large}$, $\text{FCNN}_{medium}$, $\text{FCKAN}_{medium}$, $\text{FCNN}_{small}$, and $\text{FCKAN}_{small}$.

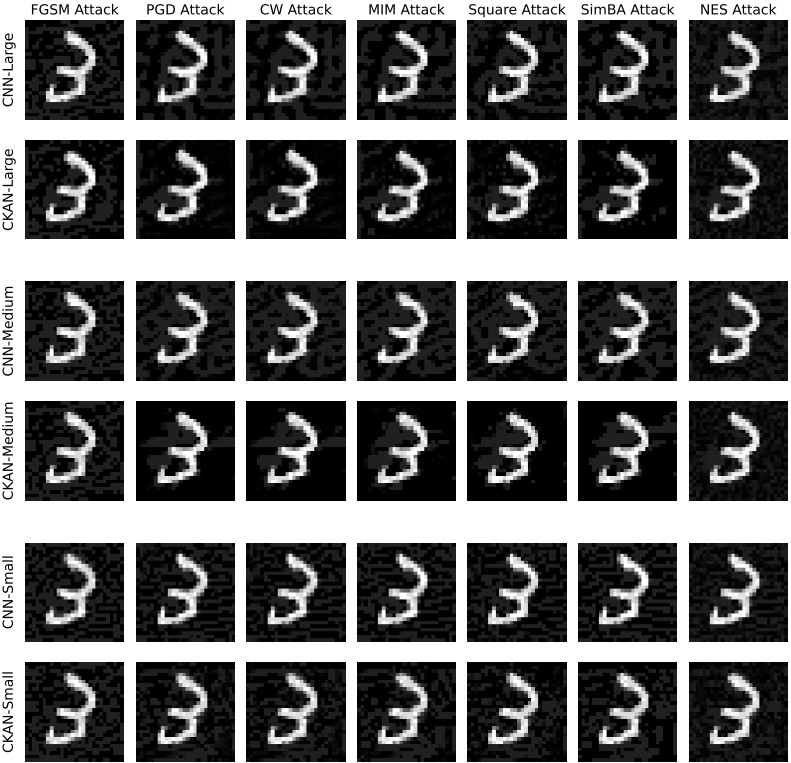

Figure 8: Visualization of the adversarial images on a single MNIST image using convolutional models. Rows represent model types and sizes: $\text{CNN}_{large}$, $\text{CKAN}_{large}$, $\text{CNN}_{medium}$, $\text{CKAN}_{medium}$, $\text{CNN}_{small}$, and $\text{CKAN}_{small}$.

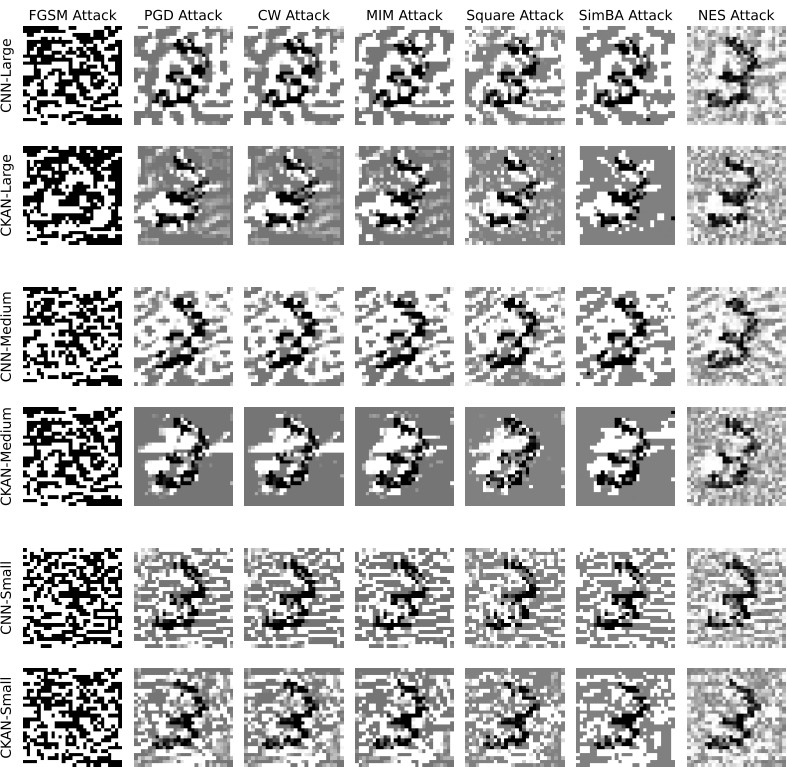

Figure 9: Visualization of the pertubations on a single MNIST image using MNIST image using convolutional models. Rows represent model types and sizes: $CNN_{large}$, $CKAN_{large}$, $CNN_{medium}$, $CKAN_{medium}$, $CNN_{small}$, and $CKAN_{small}$.

# B  Adversarial Attacks

In this appendix we provide a formal mathematical definition of the white-box and black box adversarial attacks.

## B.1  White-Box Attacks

In the context of white-box attacks, the adversary generates an adversarial example $x_{adv}$ from the original input $x$, using a specific technique, denoted as $\mathcal{A}$. The selection of $\mathcal{A}$ plays a critical role in determining the attack's success rate, which is defined as the percentage of samples that the classifier $C$ misclassifies. The literature presents a wide array of techniques for crafting white-box attacks. In the following sections, we provide a detailed description of each attack method evaluated in this study.

### B.1.1  Fast Gradient Sign Method (FGSM)

The Fast Gradient Sign Method (FGSM) (Goodfellow et al., 2014) generates adversarial examples by adding perturbations in the direction of the gradient of the loss function. Specifically, an adversarial example $x_{adv}$ is computed as follows:

$$x_{adv} = x + \epsilon \cdot \text{sign}\left(\nabla_x \mathcal{L}(x, y; \theta)\right), \tag{2}$$

where $\mathcal{L}$ is the loss function associated with a classifier parameterized by $\theta$, $\nabla_x \mathcal{L}(x, y; \theta)$ represents the gradient of the loss with respect to the input $x$, and $\epsilon$ is a small perturbation. Notably, in Equation 2, only the sign of the gradient is used, with the magnitude of the perturbation controlled by $\epsilon$. It is crucial to highlight that FGSM is a single-step attack; the adversary computes the gradient by backpropagating through the model once—and directly applies this gradient to perturb the input $x$.

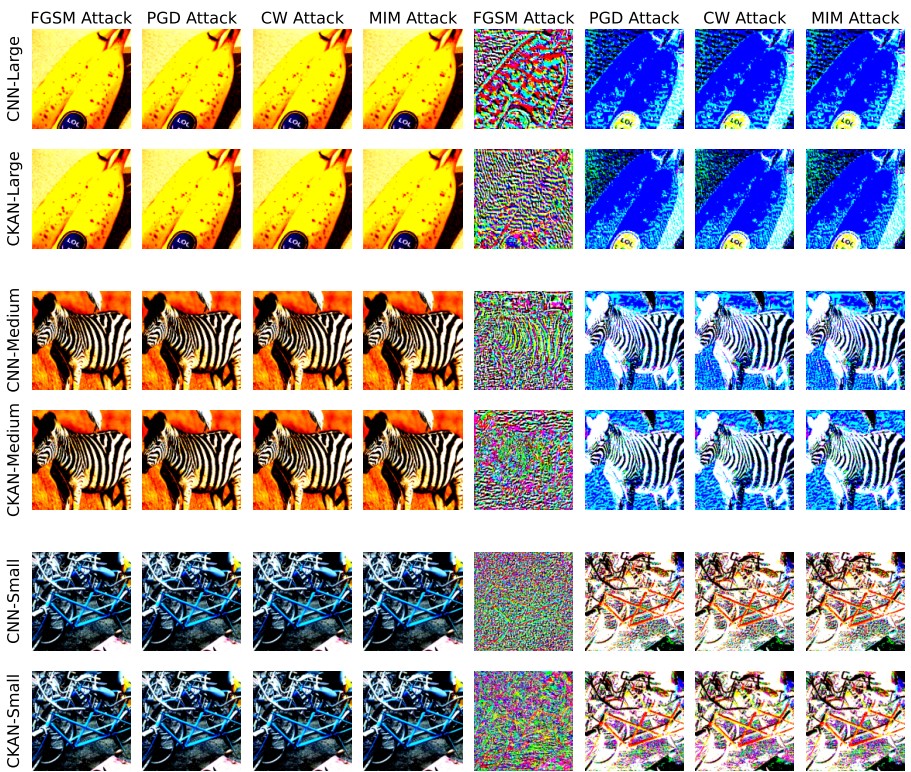

Figure 10: Visualization of the adversarial images and the perturbations generated by white-box attacks on three images from our subset of ImageNet using convolutional models. Rows represent model types and sizes: CNN$_{\text{Large}}$, CKAN$_{\text{Large}}$, CNN$_{\text{Medium}}$, CKAN$_{\text{Medium}}$, CNN$_{\text{Small}}$, and CKAN$_{\text{Small}}$. The first four images in each row represent the adversarial images, while the last four images represent the perturbations.

### B.1.2 Projected Gradient Descent (PGD)

The Projected Gradient Descent (PGD) method (Madry et al., 2017) is an iterative extension of the FGSM, designed to generate more robust adversarial examples. The PGD attack iteratively perturbs the input $x$ by repeatedly applying gradient updates and projecting the perturbed image back onto an $\epsilon$-ball around the original input. The adversarial example $x_{\text{adv}}$ after $k$ iterations is computed as:

$$x_{\text{adv}}^{k+1} = \Pi_{\mathcal{B}_\epsilon(x)} \left( x_{\text{adv}}^k + \alpha \cdot \text{sign} \left( \nabla_x \mathcal{L}(x_{\text{adv}}^k, y; \theta) \right) \right), \tag{3}$$

where $\Pi_{\mathcal{B}_\epsilon(x)}(\cdot)$ denotes the projection onto the $\epsilon$-ball centered at $x$, $\alpha$ is the step size, and $k$ indicates the iteration number.

Unlike FGSM, which performs a single gradient step, PGD iteratively refines the adversarial perturbation over multiple steps, thereby producing stronger adversarial examples. The projection step ensures that the perturbation remains within the allowable $\epsilon$-ball, maintaining the proximity of the adversarial example to the original input.

### B.1.3 Carlini & Wagner (C&W)

The Carlini & Wagner (C&W) is a powerful optimization-based adversarial attack designed to generate adversarial examples by minimizing a specially crafted loss function while ensuring that the perturbation remains imperceptible. When employing the $\ell_\infty$ norm, the attack seeks to find a minimal perturbation $\delta$ such that the resulting adversarial example $x_{\text{adv}} = x + \delta$ misleads the classifier. The optimization problem is formulated as follows:

$$\mathcal{L}(\delta, x) = \|\delta\|_\infty + c \cdot f(x + \delta), \tag{4}$$

where $\mathcal{L}(\delta, x)$ is the objective function to minimize, subject to:

$$x_{\mathrm{adv}} = \mathrm{clip}(x + \delta, 0, 1). \tag{5}$$

where $f(x_{\mathrm{adv}})$ is a loss function that is specifically designed to ensure the misclassification of the adversarial example $x_{\mathrm{adv}}$. The term $\|\delta\|_\infty$ represents the maximum perturbation applied to any individual pixel, and $c$ is a constant that balances the trade-off between minimizing the perturbation and maximizing the loss. The clipping function ensures that the perturbed image remains within the valid input space (i.e., pixel values between 0 and 1).

The attack is typically solved through an iterative optimization procedure, where the perturbation $\delta$ and the adversarial image $x_{\mathrm{adv}}$ are updated at each iteration, as follows:

$$\begin{aligned}
\delta^k &= \mathrm{clip}(x_{\mathrm{adv}}^k - x, -\epsilon, \epsilon), \\
x_{\mathrm{adv}}^{k+1} &= x_{\mathrm{adv}}^k + \alpha \cdot \mathrm{sign}\big(\nabla_x \mathcal{L}(x_{\mathrm{adv}}^k, \delta^k)\big).
\end{aligned} \tag{6}$$

where $\alpha$ is the step size, and $k$ indicates the iteration number.

### B.1.4  Momentum Iterative Method (MIM)

The Momentum Iterative Method (MIM) (Dong et al., 2018) is an enhancement of the basic iterative methods for generating adversarial examples, incorporating momentum to stabilize the gradient updates and avoid poor local maxima. The MIM attack iteratively updates the adversarial example $x_{\mathrm{adv}}$ by accumulating a momentum term, which helps to amplify gradients that consistently point in the same direction across iterations. The adversarial example $x_{\mathrm{adv}}^{k+1}$ after $k$ iterations is computed as:

$$g^{k+1} = \mu \cdot g^k + \frac{\nabla_x \mathcal{L}(x_{\mathrm{adv}}^k, y; \theta)}{\|\nabla_x \mathcal{L}(x_{\mathrm{adv}}^k, y; \theta)\|_1}, \tag{7}$$

$$x_{\mathrm{adv}}^{k+1} = \Pi_{\mathcal{B}_\epsilon(x)} \left( x_{\mathrm{adv}}^k + \alpha \cdot \mathrm{sign}\left(g^{k+1}\right) \right), \tag{8}$$

where $g^k$ denotes the accumulated gradient at iteration $k$, $\mu$ is the momentum factor, $\alpha$ is the step size, and $\Pi_{\mathcal{B}_\epsilon(x)}(\cdot)$ represents the projection onto the $\epsilon$-ball centered at $x$.

By integrating the momentum term, MIM not only improves the convergence of the attack, but also enhances its effectiveness by consistently updating the perturbation in directions that contribute most to the increase in the loss function. This makes MIM particularly potent in generating adversarial examples that can transfer across different models.

### B.2  Black-Box Attacks

In a black-box attack the adversary does not have direct access to the internal parameters or architecture of the target model. Instead, the attack relies on querying the model to gather information about its predictions or confidence scores. Adversarial examples $x_{\mathrm{adv}}$ are crafted based on these outputs, without explicit knowledge of the model's gradients or weights. Below, we describe the specific black-box attack methodologies used in this study.

### B.2.1  Transfer-based Attacks

Transfer-based attacks leverage the phenomenon of transferability, where adversarial examples generated for one model are also misclassified by other models. This concept of adversarial example transferability was first introduced by Szegedy et al. (2013).

Let $\mathcal{A}$ denote the white-box adversarial attack method, and let $f_\theta$ represent the model under attack. The attack can be formalized as:

$$\mathcal{A}_{f_\theta}(\mathbf{x}, y) = \mathbf{x}_{\mathrm{adv}}, \tag{9}$$

where $(\mathbf{x}, y)$ corresponds to the input-label pairs.

Let $g_\phi$ denote the target model, which is employed to evaluate the effectiveness of the generated adversarial examples. An adversarial example $\mathbf{x}_{\text{adv}}$ is considered to have successfully transferred via a given attack if and only if the following conditions hold:

$$g_\phi(\mathcal{A}_{f_\theta}(\mathbf{x}, y)) \neq y \quad \text{and} \quad f_\theta(\mathbf{x}) = g_\phi(\mathbf{x}) = y. \tag{10}$$

To quantify the transferability of adversarial examples between $f_\theta$ and $g_\phi$ for a specific attack, we define the transferability metric as:

$$t_{f_\theta, g_\phi}^{\mathcal{A}} = \frac{1}{m} \sum_{i=1}^{m} \begin{cases} 1 & \text{if } g_\phi(\mathcal{A}_{f_\theta}(\mathbf{x}_i, y_i)) \neq y_i \\ 0 & \text{otherwise} \end{cases}, \tag{11}$$

where $m$ is the number of samples.

To identify the most effective attack in terms of transferability between the generator model and the evaluator model, we calculated the overall transferability metric, $t_{f_\theta, g_\phi}^{\text{total}}$, as follows:

$$t_{f_\theta, g_\phi}^{\text{total}} = \max_{\mathcal{A} \in \{\text{MIM}, \text{FGSM}, \text{C\&W}, \text{PGD}\}} \left( t_{f_\theta, g_\phi}^{\mathcal{A}} \right). \tag{12}$$

### B.2.2  Square Attack

The Square Attack (Andriushchenko et al., 2020) is a query-efficient black-box adversarial attack via random search that operates by iteratively applying random perturbations to the input image. The target is to minimize the following loss:

$$L(f(x_{\text{adv}}), y) = f_y(x_{\text{adv}}) - \max_{k \neq y} f_k(x_{\text{adv}}), \tag{13}$$

At each iteration, the attack selects a random patch of the image and applies perturbations to it in the form of a square grid. The perturbations are scaled by a predefined $\epsilon$ value, ensuring that they stay within the specified $\ell_\infty$ norm constraint. The adversarial example is updated iteratively as follows:

$$x_{\text{adv}}^{k+1} = x_{\text{adv}}^k + \Delta_{\text{square}}, \tag{14}$$

where $\Delta_{\text{square}}$ represents the perturbation applied to the selected patch. The key advantages of the Square Attack are its simplicity and effectiveness, making it a strong baseline for black-box adversarial evaluation. Additionally, its random search mechanism ensures good exploration of the input space, increasing the chances of finding adversarial examples.

### B.2.3  Simple Black-Box Attack (SimBA) in DCT Space

The Simple Black-Box Attack (SimBA) (Tu et al., 2019) is a query-efficient method for crafting adversarial examples. Unlike gradient-based attacks, SimBA perturbs the input image in a random manner (e.g., DCT or Fourier space) and evaluates the effect of the perturbation on the model's confidence score. If the perturbation reduces the confidence of the correct label, it is retained; otherwise, it is discarded.

In the DCT (Discrete Cosine Transform) space, SimBA exploits the frequency-domain representation of images to prioritize perturbations on low-frequency components. This ensures that the adversarial examples remain perceptually similar to the original images while efficiently reducing the model's confidence in its predictions.

The SimBA attack updates the adversarial example iteratively as follows:

$$x_{\text{adv}}^{k+1} = x_{\text{adv}}^k + \Delta_{\text{DCT}}, \tag{15}$$

where $\Delta_{\mathrm{DCT}}$ represents the perturbation in the DCT basis. The step size of the perturbation is controlled by the $\epsilon$ parameter.

SimBA's simplicity and efficiency make it a widely-used black-box attack, particularly in scenarios where query limits are enforced.

### B.2.4   Natural Evolution Strategy (NES)

The Natural Evolution Strategy (NES) (Ilyas et al., 2018) is a gradient-free optimization method that estimates the gradient of a loss function by sampling noise vectors and using finite differences to approximate the effect of perturbations. The method is particularly suited for black-box attacks, where access to model gradients is unavailable. It iteratively updates the adversarial example by approximating the gradient of the loss with respect to the input.

For a given input $x$, the adversarial perturbation is computed by estimating the gradient using a set of $n$ random noise vectors $u_i \sim \mathcal{N}(0, I)$, sampled from a standard normal distribution. The update rule is as follows:

$$\hat{\nabla}^k = \frac{1}{2\sigma n} \sum_{i=1}^{n} \left[ \mathcal{L}(x_{\mathrm{adv}}^k + \sigma u_i, y) - \mathcal{L}(x_{\mathrm{adv}}^k - \sigma u_i, y) \right] u_i. \tag{16}$$

Using this gradient estimate $\hat{\nabla}^k$, the adversarial example $x_{\mathrm{adv}}$ is updated iteratively as follows:

$$x_{\mathrm{adv}}^{k+1} = \Pi_{\mathcal{B}_\epsilon(x)} \left( x_{\mathrm{adv}}^k + \alpha \cdot \mathrm{sign}(\hat{\nabla}^k) \right). \tag{17}$$

