# OpenReview forum: "On the Robustness of Kolmogorov-Arnold Networks: An Adversarial Perspective"
_TMLR — Accepted by TMLR_

### Review · Reviewer_YvaH · 2025-01-27

**Summary Of Contributions:**

This paper investigates the adversarial robustness of Kolmogorov-Arnold Networks (KAN) against both white-box and black-box attacks. To this end, this paper conducts a comprehensive empirical evaluation of KANs with various configurations, employing standard training methods. The results reveal that small and medium-sized KANs exhibit lower robustness compared to convolutional networks. However, the findings also indicate that larger KANs demonstrate superior robustness when compared to convolutional networks.

**Audience:**

Yes

**Broader Impact Concerns:**

No ethical concerns require attention.

**Claims And Evidence:**

No

**Requested Changes:**

The primary suggested changes are outlined in the weaknesses section.

The term safety is more appropriate than security in this context. Authors may consider using the term safety consistently throughout the paper.

**Strengths And Weaknesses:**

Strengths:

- Studying the adversarial robustness of KAN is essential for advancing the understanding in this area. The findings presented in this paper have the potential to influence future works and development efforts significantly.

- Comprehensive evaluations are highly valued and contribute significantly to the impact of the findings.

- The observation that large-scale KANs exhibit greater robustness is particularly intriguing.

---

Weaknesses:

- The evaluations in this study were conducted exclusively using standard training, which may limit the novelty of the empirical observations. While the finding that larger models exhibit greater robustness is intriguing, it is not entirely novel. Similar trends have been observed in existing benchmarks, such as RobustBench [1]. Nevertheless, the observations in this paper suggest that KANs also adhere to these established patterns.

- The differences observed in the experiments are marginal. For instance, in Table 4, the results for CW attacks, particularly given that these models were trained using standard training methods. Such findings cannot conclusively indicate that larger KANs are more adversarially robust. The reviewer suggests extending the evaluation to include adversarial training to provide more definitive insights.

- The paper lacks theoretical or empirical analysis explaining why larger KANs exhibit greater robustness. Providing such explanations would significantly enhance the contribution and deepen the understanding of the observation.

- Based on Figures 1 and 3, the results indicate that small variants of KANs demonstrate robustness comparable to the large configurations, while the mid-sized configurations are less robust. What could be the underlying reason for this pattern?

- The evaluated attacks are well-established for convolutional models. However, these attacks might overestimate the robustness of KANs. While it is understandable that only a limited range of attacks has been explored for KANs, the reviewer suggests expanding the evaluation to include ensemble-based attacks, such as AutoAttack [2].

---

[1] Croce, Francesco, et al. "RobustBench: a standardized adversarial robustness benchmark." Thirty-fifth Conference on Neural Information Processing Systems Datasets and Benchmarks Track (Round 2).

[2] Croce, Francesco, and Matthias Hein. "Reliable evaluation of adversarial robustness with an ensemble of diverse parameter-free attacks." International conference on machine learning. PMLR, 2020.

---

### Review · Reviewer_wqTz · 2025-01-30

**Summary Of Contributions:**

This paper provides an in-depth analysis of the adversarial robustness of Kolmogorov-Arnold Networks (KANs), focusing on image classification tasks. The key contributions of this work are:

- A comprehensive evaluation of KANs under adversarial conditions, including both white-box and black-box attacks such as such as Fast Gradient Sign Method (FGSM), Projected Gradient Descent (PGD), Carlini-Wagner (C&W), Momentum Iterative Method (MIM), Square Attack, Simple Black-Box Attack (SimBA), and Natural Evolution Strategies (NES).
- Comparative analysis of KANs against traditional fully connected neural networks (FCNNs) and convolutional neural networks (CNNs) across multiple benchmark datasets (MNIST, KMNIST, FashionMNIST, CIFAR-10, and SVHN).
- Experimental insights on the robustness of small, medium, and large KAN architectures, demonstrating that larger KANs exhibit superior robustness compared to their standard neural network counterparts.
- An extensive discussion on the transferability of adversarial attacks, illustrating that KANs demonstrate lower attack transferability, particularly in their larger configurations.
- A structured analysis of robustness trends, highlighting that KANs remain vulnerable to adversarial perturbations, albeit with greater resistance in larger models.

**Audience:**

Yes

**Claims And Evidence:**

Yes

**Requested Changes:**

- Provide a theoretical justification for why large KANs demonstrate superior robustness. A discussion on potential factors (e.g., lower Lipschitz constants, function decomposition) would strengthen the contributions.
- Extend the analysis to additional datasets, particularly more complex or real-world datasets, to validate the generalizability of the findings.
- Ablation studies on KAN hyperparameters (e.g., number of knots, spline order) and their impact on adversarial robustness.
- Tune attack hyperparameters specifically for the KANs and CKANs.
- Effect of adversarial training: Investigate whether adversarial training techniques can further improve KAN robustness.
- If the authors can provide some insight into provable robustness certificates for KANs, that would significantly strengthen the papers contribution.
- Visualization of adversarial perturbations, showcasing how perturbations affect KANs differently compared to FCNNs and CNNs.

**Strengths And Weaknesses:**

Strengths:
- Novelty: This is the first extensive study analyzing the adversarial robustness of KANs.
- Comprehensive Experimental Setup: The authors systematically evaluate multiple architectures (FCKANs, CKANs, FCNNs, CNNs) on multiple datasets, employing a variety of adversarial attacks.
- Strong Comparative Analysis: The inclusion of both white-box and black-box attacks ensures a thorough evaluation of adversarial robustness.


Weaknesses:
- Limited Theoretical Justification: While empirical results support the claim that large KANs are more robust, the paper lacks a strong theoretical explanation for why would KANs exhibit superior robustness. What is the motivation of KANs to be more robust than traditional neural networks? I would expect the paper should start with that some theoretical motivation (maybe from lipschitz point of view) even if this paper is purely empirical.
- Dataset Limitations: The study primarily focuses on standard image classification benchmarks. Evaluating robustness on more complex datasets, such as ImageNet, could strengthen the conclusions.
- Ablation Studies: Further investigation into hyperparameter choices (e.g., spline order, number of knots) and their impact on robustness would provide deeper insights.
- Attack settings- It is not clear if the attack settings used in this paper for KANs are optimally set. It seems they are adopted from traditional networks. For the study to be more informative, it is important for the attacks to be tuned specifically for KANs and CKANs respectively.

---

### Review · Reviewer_rVMj · 2025-01-31

**Summary Of Contributions:**

This paper presents a systematic evaluation of the robustness of Kolmogorov-Arnold Networks (KANs) against adversarial attacks in both white-box and black-box settings, using five dataset and comparing the classification accuracy with FCNNs and CNNs.

**Audience:**

Yes

**Claims And Evidence:**

Yes

**Requested Changes:**

See comments in weaknesses.

**Strengths And Weaknesses:**

Strengths:
1. The paper tests a wide range of attack methods, covering both black-box and white-box settings.
2. The paper is well-written, with clear and concise explanations, making it easy to follow.

Weaknesses:
1. The datasets used for testing are all with low resolution and relatively simple. It is necessary to include more challenging datasets, such as ImageNet for a more comprehensive evaluation.
2. Symbols that appear for the first time should be explained in the main text for clarity, such as Table 3.
3. When calculating average of attack transferability in Table 7, the case where i=j (the same model) should not be included.
4. The current analysis only involves classification accuracy without analyzing the reasons why KAN’s robustness may be better or worse than that of fully connected or convolutional networks.
5. The paper should show or discuss how to enhance the robustness of KAN.

---

### Author Response · Authors · 2025-03-12
**Camera ready revision**

Dear AE and reviewers,

We sincerely thank the editor and reviewers for their time and effort in reviewing our manuscript. Your constructive feedback has been invaluable in improving our work, and we truly appreciate your insightful suggestions. We have carefully revised the manuscript accordingly and have now uploaded the camera-ready version.

Best,

Authors.

---

### Decision · Action_Editor_k1tN · 2025-03-07

**Recommendation:** Accept as is

**Comment:**

The recommendation is based on the reviewers' comments, the action editor's evaluation, and the authors’ response.

This paper studies the (adversarial) robustness of Kolmogorov-Arnold Networks (KANs), a new family of machine learning models. The novelty in the exploration of KAN's robustness is acknowledged, and the results also provide new insights. The authors’ rebuttal has successfully addressed the major concerns of most of the reviewers. Therefore, I recommend acceptance of this submission. I also expect the authors to include the new results and suggested changes during the rebuttal phase to the final version.

**Audience:**

Of interest to researchers interested in alternative machine learning models other than neural networks

**Claims And Evidence:**

The claims of KAN's robustness are supported by the empirical results.